# A Cross-Sectional Investigation of Cardiovascular and Metabolic Biomarkers among Conventional and Organic Farmers in Thailand

**DOI:** 10.3390/ijerph15112590

**Published:** 2018-11-20

**Authors:** Pornpimol Kongtip, Noppanun Nankongnab, Mathuros Tipayamongkholgul, Ariya Bunngamchairat, Jutharak Yimsabai, Aranya Pataitiemthong, Susan Woskie

**Affiliations:** 1Department of Occupational Health and Safety, Faculty of Public Health, Mahidol University, 420/1 Rajvidhi Road, Bangkok 10400, Thailand; noppanun.nan@mahidol.ac.th (N.N.); ariya.bun@mahidol.ac.th (A.B.); 2Center of Excellence on Environmental Health and Toxicology, EHT, Bangkok 10400, Thailand; 3Department of Epidemiology, Faculty of Public Health, Mahidol University, 420/1 Rajvidhi Road, Bangkok 10400, Thailand; ajarnpuk@yahoo.com; 4Buddhachinaraj Phitsanulok, 90 Sithamma traipidok Road, Muang, Phitsanulok 65000, Thailand; jutharak@gmail.com (J.Y.); Ary.5555@hotmail.com (A.P.); 5Department of Public Health, University of Massachusetts Lowell, One University Ave, Lowell, MA 01854-2867, USA; Susan_Woskie@uml.edu

**Keywords:** cardiovascular biomarker, metabolic biomarkers, pesticide-using farmers, organic farmers

## Abstract

Pesticide exposure has been implicated as a risk factor for developing a wide range of adverse health issues. Some examples are metabolic syndromes, including diabetes. This study investigated the relationship between current occupational use of pesticides and metabolic and cardiovascular biomarker levels among organic and conventional farmers in Thailand. In total, 436 recruited farmers were divided into two groups: conventional farmers (*n* = 214) and organic farmers (*n* = 222). Participants, free of diabetes, were interviewed and submitted to a physical examination. Serum samples were collected for clinical laboratory analyses, i.e., serum glucose and lipid profiles (triglycerides, total cholesterol, high-density lipoproteins, and low-density lipoproteins). Potential risk factors such as smoking, alcohol consumption, and heavy exercise were significantly different between the two groups. There were significant differences in terms of the years of pesticide use, pesticide use at home, sources of drinking water, and distance between the farmers’ homes and farms between the groups. After adjusting for confounders, current conventional farmers had significantly higher abnormal body mass index (BMI), waist circumference, body fat percentage (% body fat), triglyceride, total cholesterol, and low-density lipoprotein values as compared to organic farmers. Conventional farmers had higher risk of many metabolic and cardiovascular risk factors as compared to organic farmers, putting them at higher risk of metabolic diseases in the future.

## 1. Introduction

Pesticides are widely used in agriculture with a goal of enhancing productivity, which, in turn, can improve nutrition for the 795 million people experiencing food insecurity worldwide [1]. Nonetheless, the extensive use of pesticides globally has also triggered growing concerns regarding environmental impacts and adverse health effects [2,3,4]. Several organochlorine pesticides were associated with metabolic syndrome [5]. Approximately 105 pesticides have been identified as endocrine disruptors affecting the natural hormone balance in the human body [6]. Accumulating evidence suggests that endocrine disrupting chemicals can contribute to the development of diverse chronic diseases, including metabolic disorders such as diabetes [5]. Thailand ranks fourth in annual pesticide consumption worldwide [7] and in 2017, Thailand imported 198,317 tons of pesticides, including 148,979 tons of herbicides, 21,601 tons of insecticides, and 19,923 tons of fungicides [8]. Importation and domestic production continue to increase, despite efforts by the Thai government to restrict the import of some pesticides, and proposals for new agrochemical regulations. 

Since the 1990s the use of organic farming has been gradually increasing in Thailand [9]. Driven by factors related to government initiatives to encourage sustainable agriculture in line with King Rama the Ninth’s self-sufficiency policy, as well as market demands, the use of organic farming practices is growing [10]. At present, Thailand is ranked 60th in the world and the eighth in Asia in terms of organic agricultural area (45,587 hectares), which, in 2015, represented only 0.2% of the total agricultural area in Thailand [11]. Rice was the major crop grown organically, occupying 60% of the organic land area, followed by other cash crops such as sugarcane, peanuts, and corn (20%), and diverse kinds of vegetables and fruits (4%) [10]. 

As part of a longitudinal study of the non-communicable disease impacts of occupational pesticide exposures, we examined the baseline cross-sectional data on the prevalence of adverse biomarkers that are risk factors for the development of metabolic diseases such as diabetes, stroke and cardiovascular disease. We focused this analysis on the comparison of conventional farmers who currently use pesticides with organic farmers who do not currently use pesticides in order to investigate the role of current pesticide use plays in the risk of developing adverse metabolic biomarkers.

## 2. Materials and Methods 

### 2.1. Study Population and Data Collection 

We collected this cross-sectional data as the baseline for a prospective follow-up study. To ensure a variety of pesticide exposures were included, as well as organic farmers, we recruited farmers from three provinces in Thailand. For agriculture involving use of chemical pesticides, we recruited rice and vegetable farmers in two sub-districts of the Phitsanulok province, which is located in the lower north of Thailand. Conventional farmers who grew sugarcane (59%), rice (20%), cassava (16%), and maize (2%) were recruited from the Nakorn Sawan province, which is situated in the upper part of the central area. One farmer who sprayed pesticides was selected in each household. To recruit organic farmers, five sub-districts in the Yasothorn province, the large organic farming area in Thailand were selected. All organic farmers had to use certified organic farming practices for all of their crops.

The study included male or female farmers over 18 years old who were free of a current diagnosis of diabetes, high blood pressure, and thyroid or heart disease. Health-promoting hospitals/primary care units (HPH/PCUs) were the focal point in coordinating with agricultural workers and community leaders in the designated study areas. We hired a site officer from the local community to work with the HPH/PCU and community leaders in each area to recruit subjects. The site officer visited each home for recruitment and to conduct interviews. The subjects received a small remuneration for participation in this study and received the individual health check-up results explained by a study doctor. At recruitment we enrolled 243 conventional farmers and 235 organic farmers. Eight months later, when the first physical health checkups were conducted, the cohort declined to 214 conventional farmers and 222 organic farmers, with response rates of 88.1 and 94.5%, respectively. This study was approved by the Faculty of Public Health, Mahidol University Institutional Review Board (MUPH 2015-146). The questionnaires comprised screening questions, farmer characteristics, health behaviors, family health history, food consumption behaviors, home and demographic information, self-reported health problems, agricultural activities, and history of pesticide use. The Department of Mental Health stress questionnaire (Thai Ministry of Health) [12] was used and covered sleeping problems, reduced concentration, irritability, boredom, and lack of interest in meeting people, with a scale of 0 for no problem to 3 as always having problems. Data on weight, height, waist circumference, body composition (Tanita model DC-360, Amsterdam, Netherland), blood pressure (taken twice, 10 min apart and averaged), and blood samples were obtained by physical examination. Sera extracted from blood samples in non-heparinized vacutainer tubes were stored at −20 °C until analysis. Serum glucose, triglycerides (TGs), total cholesterol (TC), high-density lipoprotein (HDL), and low-density lipoprotein (LDL) were measured by an AU5800 apparatus (Beckmann Coulter, Atlanta, Georgia, USA) at Buddhachinaraj Hospital, the regional medical center in the Phitsanulok province, using standard clinical laboratory methods.

All subjects gave their informed consent for inclusion before they participated in the study. The study was conducted in accordance with the Declaration of Helsinki, and the protocol was approved by the Ethics Committee of Human Research, Faculty of Public Health, Mahidol University (MUPH 2015-146).

### 2.2. Study Variables

The study variables were defined according to internationally recognized clinical criteria [13]. Normal blood sugar was defined as ≤125 mg/dL. As for normal lipid profiles, they were defined as follows: total cholesterol (TC) of ≤200 mg/dL; low-density lipoprotein (LDL) of ≤100 mg/dL; and high-density lipoprotein (HDL) above or equal to 60 mg/dL. The clinical criteria for high blood pressure used the Thai guidelines on the treatment of hypertension (normal values of <140 and <90 mmHg for systolic and diastolic blood pressure) [14]. BMI was classified according to the WHO cut-off points of <18.50–24.99 kg/m^2^ (normal range), and ≥25 (overweight) [15]. The body composition fat % was classified as normal (male ≤ 27%; female ≤ 25%) and abnormal (male > 27%; female > 25%) [16]. Metabolic syndrome was defined as the presence of three or more of these risk factors: (1) Central obesity, measured by waist circumference, for Thai people, of >90 cm (35 in) in men or >80 cm (32 in) in women; (2) fasting blood triglycerides of >150 mg/dL (1.7 mmol/L); (3) HDL cholesterol <40 mg/dL (1.04 mmol/L) in men or <50 mg/dL (5.6 mmol/L) in women; (4) elevated blood pressure of 130/85 mmHg or higher; and (5) fasting blood glucose >100 mg/dL (5.6 mmol/L) [13].

### 2.3. Statistical Analyses 

All statistical analyses were done by SPSS for Windows, Version 18.0. (SPSS (Thailand) Co., Ltd., Bangkok, Thailand). When comparing conventional and organic farmers, contingency tables, Chi squared, Fisher’s exact test, and independent *t*-tests were employed. Due to the high prevalence of the health outcomes we used a Poisson generalized log-linear model to investigate the factors that increased the risk of an abnormal biomarker for metabolic-related disease. All demographic and behavioral risk factors were examined as covariates in univariate models with each of the health outcome parameters. If the variable showed significance in a number of univariate analyses, it was included in the final model. For pesticide exposure, we used the farming group (conventional vs. organic) to represent current pesticide use as well as variables for historical pesticide use (categories) and current use of insecticides at home (yes/no).

## 3. Results

### 3.1. Characteristics of Agricultural Workers

We recruited 436 Thai farmers to participate in this study. Forty-nine percent (*n* = 216) were conventional farmers and fifty-one percent (*n* = 222) were organic farmers. Demographic characteristics of the conventional and organic farmers are found in Table 1. 

Conventional farmers were significantly more likely to be male, had a lower average age, and were less well educated than organic farmers. There was no significant difference between the farming groups in terms of average hours of agricultural work per day, marital status, or income (Table 1). Second jobs reported by conventional farmers involved general labor (26%), grocery stores (8%), construction (6%), garage work (6%), etc. The second jobs of organic farmers involved general labor (25.8%), or employment as merchants (8.9%), construction workers (8.1%), government officers (3.4%), etc. 

### 3.2. Risk Factors 

In looking at the main behavioral risk factors for metabolic syndrome reported by conventional farmers and organic farmers, we found a significant difference in alcohol use, smoking, and exercise patterns (Table 2).

Farmers were asked to self-assess their stress by scoring the frequency with which they experienced difficulty sleeping, reduced concentration, irritation, anxiety, boredom, and introversion. The number of farmers who reported almost never having these symptoms in the past 2–4 weeks was not significantly different between organic and conventional farmers (*p* = 0.084). The organic farmers who participated in this study were certified organic farmers who were not allowed to use any synthetic chemicals for the whole process of growing and harvesting crops. However, most of the organic farmers had used pesticides before becoming organic farmers. Only 29 organic farmers (12.7%) reported never having sprayed pesticides. The average number of years of pesticide use by the conventional farmers was 26.9 years, ranging from 4 to 51 years, while the average number of years of pesticide use by the organic farmers, prior to becoming organic farmers, was 16 (ranging from 0 to 40 years). Conventional farmers reported using several types of pesticides depending on the types of crops, insects, and weeds (Table 3). Regarding distances from the house to the farm, most conventional farmers (84.6%) lived closed to their farm (within 1 km), whereas significantly fewer organic farmers resided near their farm (46.8%). For drinking water, most of the organic farmers (65.1%) drank bottled water, while only 6.5% of conventional farmers drank bottled water (*p* < 0.001). 

### 3.3. Food Intake Behavior of Chemical Use and Organic Farmers in the Past Month

Self-reported food intake in the past month was classified as 5–7 days per week versus less often (Table 4). There was no significant difference in the intake of vegetables with five colors, or vegetable and fruit intake of <0.5 kilograms per day between organic and conventional farmers. Organic farmers had significantly higher consumption of sweet fruits and desserts than those of conventional farmers. Frequent intake of sweet drink beverages, meat with high fat, and deep fried food consumption were not significantly different between the two groups, although 22–61% reported consuming these on 5–7 days per week. For snacks or salty/ preserved foods such as pickles, salt fish, instant noodles, or porridge, the organic farmers had significantly higher consumption than the conventional farmers.

### 3.4. Comparison of Chemical Use and Organic Farmers for Health Outcomes

There were significantly more conventional farmers with an abnormal BMI than organic farmers (Table 5). The percentages of subjects with central obesity, as measured by abnormal waist circumference and % body fat, were not significantly different between the conventional and organic farmers. With regards to abnormal lipid profiles, although abnormal triglyceride levels were not significantly different between conventional and organic farmers, the percentages of subjects with abnormal total cholesterol and low-density lipoprotein (LDL) levels were significantly higher among the conventional farmers, as compared to the organic farmers. However, the percentage of subjects with abnormally low levels of high-density lipoproteins (HDLs) was significantly higher among organic farmers than conventional farmers. 

After adjusting for demographic variables (age, gender), behavioral risk factors (alcohol use, smoking, heavy exercise, eating less than half a kilogram of fruits and vegetables 5–7 days/week), and pesticide use patterns (history and at home use), the relative risk for most health outcomes/biomarkers was higher among conventional vs. organic farmers increased with the exception of blood glucose, blood pressure, and metabolic syndrome. Abnormal HDL levels were significantly lower among conventional farmers (Table 6). 

## 4. Discussion

High pesticide residues have been found in fruits and vegetables in Thai food markets and are a current public health concern [4,17,18]. As a result, many Thai people are increasingly choosing organic food. The Thai Pesticide Alert Network (Thai-PAN) tested 296 samples of fruits and vegetables from five areas of Thailand in 2016 and found that 51.4% of the fruits and vegetables granted the “Q (quality) mark” by the Thai National Bureau of Agricultural Commodity and Food Standards were contaminated with pesticide residues above the maximum residue limit (MRL) [19].

Most of the organic farmers in this study (86.9%) had previously been pesticide-using farmers before recognizing the toxicity of the pesticides they used and switching to organic farming. As a result, they were, on average, older than the conventional farmers and also had higher education levels. This finding is similar to that of a study by Chouichom (2010) that reported longer-term farm experience (higher age) and higher education were supporting factors in the adoption of an organic farming system [20,21].

The conventional farmers were more likely to be male, probably because the inclusion criteria specified that the conventional had to spray pesticides and most of sprayers in Thailand are male. This may also explain the higher use of alcohol and smoking among conventional farmers, as these behaviors are more typical among Thai males. The prevalence rate of smoking in Thailand was 41.7% in males and 1.9% in females in 2007 [22]. Females aged 15–18 years old and those over 60 years old had the lowest and highest smoking rates, at 0.1% and 3.6%, respectively [22]. Males aged 15–18 years old and 41–59 years old had the lowest and highest rates, at 14.2% and 47.5% respectively [22]. However, another explanation for the difference in smoking and alcohol use may be that training in organic farming emphasizes promotion of a better quality of life [23]. Almost 48% of organic farmers regularly undertook heavy exercise. Regular physical exercise has been documented for primary and secondary prevention of several chronic diseases (e.g., cardiovascular disease, diabetes, cancer, hypertension, obesity, depression) and premature death [24].

Regarding stress, both groups encountered similar stress problems, which could be due to economic difficulties. Approximately 35% of conventional and organic farmers were in debt and 57% of the organic farmers had a second job in addition to agricultural work. In 2013, of the 4.8 million people living in rural areas where most are working in agriculture, 10.9% lived in poverty. The poverty line in 2015 was USD83/month [25]. Organic farming in Thailand is still at an early stage where the majority of the organic products are rice and fresh vegetables [26]. At present, the Royal Project Foundation, a government program, has helped farmers to overcome the initial constraints to conversion from conventional to organic farming. Nonetheless, organic farmers still encounter concerns as price stability, product marketing, and the need for continued government support [27]. 

Most conventional farmers (85%) lived within 1 km of their farms which could result in greater opportunity for pesticide spray drift and take-home pesticide exposures. Most of the organic farmers (65.1%) drank bottled water, while most conventional farmers drank filtered tap water. Pesticide contamination levels in commercial bottled water or government-provided tap water have not been reported. In addition, a small percentage of both conventional (5.1%) and organic (2.7%) farmers reported drinking well water, where the pesticide contamination of the ground water has not been assessed. The selection of drinking water depends on availability, affordability and safety perception. Eighty-nine percent of the pesticide using farmers reported home use of insecticides to get rid of mosquitoes and other insects. Mosquito repellants may contain insecticides depending on the brand used. For example, most mosquito coils contain pyrethroids. 

In this study we found a prevalence of metabolic syndrome of 38% among conventional farmers and 40% among organic farmers. This compares to the working population Thai professionals and office workers in Bangkok where the overall prevalence was 15.2% and was approximately three times higher in men than women (25.8% vs. 8.2%). Among this “white collar” population, the most common abnormalities in men were high blood pressure (45.0%), BMI >25 kg/m^2^ (40.7%), and hypertriglyceridemia (38.7%) [28]. Among women they were high blood pressure (22.8%), BMI > 25 kg/m^2^ (20.9%), and low HDL-cholesterol (18.4%). In the general Thai population, the prevalence of metabolic syndrome was 23.2% for adults aged ≥20 years (19.5% in men and 26.8% in women) from the fourth Thai National Health Examination Survey (NHES IV) 2008–2009. In rural areas, men who had metabolic syndrome were older and more obese, with higher levels of blood pressure, fasting plasma glucose, and triglycerides but lower levels of HDL and education than those who did not [29]. In this current study, 74% of organic farmers had BMI in the normal range, which was significantly greater than the conventional farmers (58%). However, there was no significant difference between the percentages of conventional and organic farmers who had an abnormal % body fat or an abnormal waist circumference. This may be because the criteria for normal and abnormal % body fat and waist circumference take into account gender, while the criteria for abnormal BMI does not account for gender. Because there were more men in the conventional farmer group, and men tend to have more muscle mass, their BMI may be higher and fall into the “abnormal” range even though their % body fat or waist circumference do not. Another explanation for the difference in BMI could be that organic farming is more labor-intensive, while chemical farmers use more agricultural machinery in their operations [20]. However, use of insecticides [30] or other persistent organic pollutants (POPs) [31] and endocrine disrupting chemicals (EDC) [32], like those used by conventional farmers, have previously been linked to increased risk of obesity. Pesticides may cause obesity by altering homeostatic metabolic set-points, disrupting appetite controls, perturbing lipid homeostasis to promote adipocyte hypertrophy, or stimulating adipogenic pathways that enhance adipocyte hyperplasia during development or in adults [33,34].

Conventional farmers were significantly more likely to have abnormally high biomarker levels for TC and LDL, but organic farmers were more likely to have abnormally low biomarker levels for HDL. Abnormally high TC and low HDL are two risk factors for metabolic syndrome. This may explain why there was no significant difference in metabolic syndrome between these two farming types. The prevalence of metabolic syndrome in the current study (38.2–40.1%) is considerably lower than the 60% in the Anniston Health Study sample [35] carried out in agricultural workers, because our study recruited subjects free of a current diagnosis of diabetes, high blood pressure, and thyroid or heart disease. In this study there was no significant differences in abnormal blood pressure between conventional and organic farmers. However, Samsuddin et al. (2016) found that mosquito control workers with chronic mixed pesticide exposure had significantly higher brachial and aortic diastolic and systolic blood pressure levels compared to the control group [36]. POP exposure may contribute to excess adiposity and other forms of metabolism dysfunction [31]. Organochlorine pesticide exposure was significantly associated with the risk of metabolic syndrome in the Anniston Health Study population, while no association was observed for PCBs [5].

Organic and conventional farmers differed in a variety of characteristics, including education, age, gender, alcohol and smoking behavior, exercise, water consumption patterns, food intake, pesticide use at home, and years of pesticide use. Once these factors were controlled for in multivariate models for the risk of an abnormal health outcome/biomarker level, there still remained an association between farming group and most health outcomes. This suggests that current pesticide use (near-term exposure) may be an important risk factor in altering metabolic biomarkers. This finding should be replicated in other contexts to ensure that it is generalizable to other populations of farmers currently using pesticides versus those who do not. In the Dominican Republic, a study of pesticide-using vs. organic farmers found significantly higher rates of nuclear anomalies among the pesticide using farmers compared to non-exposed organic farmers, suggesting the pesticide farmers could be at higher risk for developing cancer [37]. In addition, pesticide-using farmers had significantly increased symptom frequencies related to symptoms of neurotoxicity, parasympathic effects, and acetylcholine esterase inhibition compared to organic farmers [38]. The biology of these effects are not known, but there is increasing evidence linking exposure to endocrine disrupting chemicals (EDCs) with obesity and metabolic syndrome in animals, especially when exposures occur early in life [32]. For diabetes, there is growing evidence from human studies linking exposure to EDCs with type 1 and 2 diabetes [39,40]. In a meta-analysis of 14 epidemiologic studies, there was an increased risk of coronary heart disease among those with subclinical hypothyroidism (normal thyroxine (T4) with high thyroid stimulating hormone (TSH)) [41]. Triglycerides, TC, and LDLs have been shown to increase with increasing TSH levels, and HDL decrease with increasing TSH levels [42]. In addition, as TSH levels increase, both systolic and diastolic blood pressure increase [43]. 

The strength of this study is that it is the first to compare organic and conventional farmers’ health outcomes and the first in Thailand to examine metabolic syndrome risk factors and outcomes among Thai farmers. However, a limitation of the study is its cross-sectional approach, which makes it difficult to propose causal associations. The comparison of current pesticide uses versus those who do not currently use pesticides (organic) was confounded by a number of demographic and behavior factors that differed between the two groups. Thus, we cannot be sure that even after including a number of these factors in our final model, that the designation of organic versus conventional is not a surrogate for additional uncontrolled behavioral risk factors rather than the current use of pesticides. Future work will include evaluation of the longitudinal differences in health outcomes in these two groups as well as studies of the impact of acute pesticide exposures on some of the biomarkers evaluated here. 

## 5. Conclusions

Conventional farmers who currently use pesticides had a significantly higher risk of abnormal metabolic and cardiovascular health outcome/biomarkers, including abnormal BMI, waist circumference, % body fat, triglycerides, total cholesterol, and LDL as compared to organic farmers, after controlling for a variety of demographic and behavioral risk factors. Although we cannot definitively link pesticide use with these health outcomes, conventional farmers who use pesticides are likely to be at a higher risk of metabolic diseases in the future than organic farmers.

## Figures and Tables

**Table 1 ijerph-15-02590-t001:** Demographic characteristic of conventional farmers (*n* = 214) and organic farmers (*n* = 222).

Variables	Conventional Farmers *n* (%)	Organic Farmers *n* (%)	* *p*-Value
**Age**			
<35 years	23 (10.7)	6 (2.7)	0.002
35–60 years	155 (72.4)	165 (74.7)	
>60 years	36 (16.8)	50 (22.6)	
Min-max	18–69	28–79	
Mean(SD)	50.22 (11.0)	53.11 (10.4)	0.005
**Sex**			
Male	159 (74.3)	114 (54.1)	<0.001
Female	55 (25.7)	108 (48.5)	
**Educational level**			
Below elementary	14 (6.6)	4 (1.8)	0.033
Elementary	122 (57.3)	119 (54.6)	
High school	72 (33.8)	84 (38.5)	
Bachelors or higher	5 (2.3)	11 (5)	
**Marital status**			
Single	21 (10)	13 (6)	0.067
Married	179 (85.6)	183 (85.1)	
Widowed/divorced	9 (4.3)	19 (8.8)	
**Expense adequacy**			
Enough for saving	39 (18.2)	49 (22.5)	0.420
Just enough	100 (46.7)	93 (41.3)	
In debt	75 (35)	77 (36.2)	
**Agricultural work time (hours/week)**			
Mean(SD)	26.8 (13.8)	28.7 (17.3)	0.248
**Have Second Job**			
Number(%)	50 (23.6)	124 (56.6)	<0.001
**Second job work time (hours/week)**			
Mean (SD)	25.2 (13.7)	25.4 (17.4)	0.933
**Main types of plant crop**			
Rice	163 (76.2)	211 (96.8)	
Sugarcane	80 (37.4)	14 (6.7)	
Banana	14 (6.5)	117 (56)	
Vegetables	81 (37.9)	177 (84.3)	
Other fruit	11 (5.1)	101 (48.6)	

* *p* from *t*-test or χ^2^ and independent *t* test.

**Table 2 ijerph-15-02590-t002:** Risk factors between conventional farmers (*n* = 214) and organic farmers (*n* = 222).

Risk Factors	Conventional Farmers *n* (%)	Organic Farmers *n* (%)	** p*-Value
**Alcohol intake**			
Current drinker	137 (64)	90 (41)	<0.001
Former drinker	7 (3.3)	45 (20.5)	
Non drinker	70 (32.7)	85 (38.6)	
**Smoking**			
Current smoker	58 (27.2)	35 (15.8)	0.002
Former smoker	10 (4.7)	25 (11.3)	
Non smoker	145 (68.1)	161 (72.9)	
**Heavy exercise in past month**			
Yes	59 (27.6)	102 (47.7)	<0.001
No	155 (72.4)	112 (52.3)	
**Duration of heavy exercise**			
>20 min	48 (81.4)	86 (84.3)	0.628
<20 min	11 (18.6)	16 (15.7)	
**Days/week heavy exercise**			
>3 days/week	34 (58.6)	57 (73.1)	0.076
<3 days/week	24 (41.4)	21 (26.9)	
**Stress in past 2–4 weeks**			
Yes	115 (53.7)	100 (45.5)	0.084
Almost Never	99 (46.3)	120 (54.5)	
**Years of pesticide use**			
0 years	0 (0)	29 (13.1)	
1–10 years	46 (21.9)	61 (27.6)	
11–20 years	45 (21.4)	76 (34.4)	<0.001
21–30 years	42 (20)	32 (14.5)	
>30 years	77 (36.7)	23 (10.4)	
Average (years)	26.9	16	
Range (years)	4–51	0–40	
**Living near farm**			
Yes	181 (84.6)	103 (46.8)	<0.001
No	33 (15.4)	117 (53.1)	
**Water consumption source**			
House and community well	11 (5.1)	6 (2.7)	
Bottled water	14 (6.5)	142 (65.1)	<0.001
Rain water	6 (2.8)	64 (28.4)	
Tap water	183 (85.5)	8 (3.7)	
**Insecticide use in home**			
Yes	191 (89.3)	32 (14.5)	<0.001
No	23 (10.7)	189 (85.5)	

* *p* from χ^2^ and Fisher’s exact test.

**Table 3 ijerph-15-02590-t003:** The common pesticides used by the conventional farmers in this study.

Insecticide	Herbicide	Fungicide
Chlorpyrifos	Glyphosate	Azoxystrobin
Methyl parathion	Paraquat	Difenoconazole
Carbofuran	2,2-D sodium salt	Propiconazole
Carbosulfan	Diuron	Propineb
Cypermethrin	Amethrin	carbendazim
Abamectin	Acetochlor	Isoprothiolane
Profenofos	Propanil	Mancozeb

**Table 4 ijerph-15-02590-t004:** Conventional (*n* = 214) and organic farmers (*n* = 222)’ diet for 5–7 days per week in the prior month.

Food Intake (% Reporting Intake 5–7 Days Per Week)	Conventional Farmers *n* (%)	Organic Farmers *n* (%)	* *p*-Value
Vegetables with five colors	75 (35.0)	87 (39.5)	0.333
Vegetable and fruits <0.5 kilogram	129 (60.3)	118 (53.9)	0.179
Sweet fruit (ripe mango, durian, jackfruit)	19 (8.9)	77 (35.0)	<0.001
Sweet beverages (soda, juice, milk tea, sweet coffee)	131 (61.2)	119 (53.8)	0.120
Meat with high fat (pork belly, chicken skin, grilled pork neck, squid, crab, mussels, animal intestines)	44 (20.6)	50 (22.8)	0.567
Deep-fried food (fried chicken, deep fried banana, chips)	46 (21.5)	50 (23.0)	0.700
Instant or preserved food (pickles, salt fish, instant noodles, or porridge)	20 (9.3)	93 (42.5)	<0.001
Any snack (crispy fish, potato chips, crispy seaweed)	11 (5.1)	32 (14.6)	0.001
Dessert (cake, Thai dessert (custard), bread, cookie, bakery goods)	6 (2.8)	29 (13.2)	<0.001

* *p* from χ^2^.

**Table 5 ijerph-15-02590-t005:** Comparison of conventional (*n* = 214) and organic farmers (*n* = 222) for health outcomes using clinical criteria.

Health Outcomes	Conventional Farmers *n* (%)	Organic Farmers *n* (%)	* *p*-Value
**BMI (kg/m^2^)**			
Normal (<18.49–24.99 )	123 (57.7)	165 (74.3)	<0.001
Overweight (>25.00)	90 (42.3)	57 (25.7)	
**Waist circumference (cm)**			
Normal (male ≤ 90, female ≤ 80)	137 (64.0)	148 (66.7)	0.561
Abnormal (male > 90, female > 80)	77 (36.0)	74 (33.3)	
**% Body Fat (%)**			
Normal (male ≤ 27; female ≤ 25)	113 (53.1)	107 (48.2)	0.311
Abnormal (male > 27; female > 25)	100 (46.9)	115 (51.8)	
**Triglycerides (mg/dL)**			
Normal (≤150)	124 (58.2)	143 (64.4)	0.184
Abnormal (>150)	89 (41.8)	79 (35.6)	
**Total cholesterol (mg/dL)**			
Normal (≤200)	50 (23.5)	123 (55.4)	<0.001
Abnormal (>200)	163 (76.5)	99 (44.6)	
**HDL (mg/dL)**			
Normal (≤60 mg/dL)	57 (26.8)	15 (6.8)	<0.001
Abnormal (<60 mg/dL)	156 (73.2)	207 (93.2)	
**LDL (mg/dL)**			
Normal (≤100)	23 (10.6)	65 (28.3)	<0.001
Abnormal (>100)	193 (89.4)	165 (71.7)	
**Blood glucose (mg/dL)**			
Normal (≤125)	185 (86.4)	202 (91.0)	0.133
Abnormal(>126)	29 (13.6)	20 (9.0)	
**Blood Pressure (mmHg)**			
Normal (<140 and <90)	125 (61.3)	142 (68.9)	0.104
Abnormal (≥140 and ≥90)	79 (38.7)	64 (31.1)	
**Metabolic syndrome** **			
Non-metabolic syndrome	131 (61.8)	133 (59.9)	0.688
Metabolic syndrome	81 (38.2)	89 (40.1)	

* *p* from χ^2^. ** metabolic syndrome is the presence of three or more of these risk factors: (1) Abnormal BMI; (2) abnormal blood triglycerides; (3) abnormal HDL cholesterol; (4) elevated blood pressure; and (5) abnormal blood glucose. BMI: body mass index; HDL: high-density lipoprotein; LDL: low-density lipoprotein.

**Table 6 ijerph-15-02590-t006:** Comparison of health outcomes between conventional (*n* = 214) and organic farmers (*n* = 222) using a generalized linear model for probability of abnormal blood test results.

Health Outcome	Conventional and Organic Farmers	* *p*-Value
	Adjusted ^a^ RR (95%CI) ^b^	
BMI	1.83 (1.20–2.78)	0.005
Waist circumference	1.69 (1.13–2.51)	0.010
% body fat	1.31 (1.05–1.64)	0.018
Triglyceride	1.51 (1.01–2.27)	0.045
Cholesterol	2.20 (1.69–2.86)	<0.001
HDL	0.83 (0.73–0.95)	0.008
LDL	1.34 (1.14–1.57)	<0.001
Blood glucose	1.54 (0.67–3.53)	0.309
Blood pressure	1.54 (0.98–2.42)	0.064
Metabolic syndrome	1.42 (0.94–2.13)	0.094

^a^ Adjusted for sex, age, smoking, drinking alcohol, eating less than half a kilogram of fruits and vegetables 5–7 days/week, heavy exercise, history of pesticide use, and insecticide use at home. ^b^ RR (95% CI) referred to relative risk (95% confident limit). * *p*-Value from generalized linear model.

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
