# Peer review of "A Cross-Sectional Investigation of Cardiovascular and Metabolic Biomarkers among Conventional and Organic Farmers in Thailand"

_ijerph, 2018, doi:10.3390/ijerph15112590_

Round 1
Reviewer 1 Report
This is a cross-sectional study comparing 2 groups of Thai farmers: pesticide sprayers versus organic farmers. The study investigates differences in metabolic parameters (glucose and lipid levels, blood pressure, BMI, body fat distribution). This is a valid approach and the authors promise that they will follow up the sample. So although this is a cross-sectional study there are further (cohort study) results to be expected in the future.
The authors investigate a long list of outcomes defined as binary (normal versus pathological value). I wonder why they not also investigated the effects on a continuous basis as originally they do have continuous data. I do agree that an increase in the frequency of pathological values is relevant. But often with continuous values the study power is increased and it is also interesting to quantify the size of an effect. Studying and discussing only binary values also led the authors to some mistakes. E.g. on page 9, line 271/272 they claim: “Pesticide using farmers had significantly higher levels of triglycerides, TC and LDL but lower HDL than organic farmers.” The latter statement is not true. Pesticide farmers do have a lower number of pathological values of HDL but that would mean they have higher HDL levels.
The 2 groups of farmers differ in many instances like age, sex, education, and alcohol and tobacco consumption. The authors do correct for these confounders in their final model. In general sprayers display more unhealthy features (alcohol and tobacco, exercise) and have more (additional) indicators of pesticide exposure (years of use, distance to farm, use at home). One would assume that these factors would by their own increase the risk of metabolic pathologies. So when controlling for these factors we would expect the remaining OR to be lower. But in fact table 5 demonstrates stronger effects after adjustment. The reader might wonder which factor was driving this effect of adjustment. My most likely bet would be age. But this remains to be demonstrated.
I would expect pesticide exposure to affect metabolic pathologies through chronic exposure. Therefore I would expect stronger effect with duration of exposure compared to current exposure which is the only exposure indicator for which results are reported. But the authors even included years of exposure in their models as a confounder. It would be interesting to learn about the impact of this exposure indicator as well.
Minor details:
The authors describe the sprayers by several different terms: pesticide exposed farmers, chemical use farmers, chemical farmers,…. The latter term is problematic. But also “chemical use” is not very nice because in the end “everything” is a chemical, even water (H2O). Why not define a single term and then use it throughout the paper. I now used “sprayers” but the authors also could use “conventional farmers” or whatever term they prefer.
Tables are good to report many figures. It is not necessary to report all the figures again in the text. For example the findings reported in table 1 are again reported in detail on page 4, lines 118ff. It would suffice only to report the main highlights. The details on second jobs and the types of plants grown could also go into the table instead of the text. Similar considerations apply for the risk factors (table 2). Especially the description of drinking water sources gets confusing and does not really seem necessary.
Table 4 and table 5 (unadjusted part) are redundant.
Language needs improvement. One example (apart from the term “chemical farmer”) is use of plural/singular forms, e.g. lines 18 (exposure – factors), 19 (there is only one metabolic syndrome), 123-124 (hours was). Some terms are not very usual like “expense adequacy” (table 1): why not “income”? When you classify farmers into age groups (line 119ff) you do not show that the sprayers are “younger” but that there are “more young farmers” among that group. I do not understand why raising animals or growing plants (lines 127f) is not a farmer’s job.
Line 105: I am sure this was not a Poisson regression but most likely a logistic regression because the outcome variables are binary.
Line 176: two times organic farmers.
I would not classify “no pesticide use ever” and up to 10 years of pesticide use in the same category (table 2).
Line 58ff: (also stated later in the discussion): it is simply not true that this is the first study comparing exposed to unexposed farmers. Even this journal only recently published 2 papers from Dom Rep where organic and conventional farmers were compared. Strengths and weaknesses of a paper are usually discussed in the discussion section, not advertised in the introduction.
Line 72: what means “second highest organic farming area”? Percentage or area size?
Line 74: “males or female farmers“: this text could be improved.
Description of table 5: Odds ratios of pathological values in conventional farmers compared to organic farmers. If you only write “comparison between” you could also mean the ORs for organic farming (compared to the other group). In the methods section the authors claimed they did a Poisson regression. Here they state it is a GLM which indeed would allow for several families of models. “Poisson” would be wrong and “GLM” is not very informative. And, by the way, the results are not exclusively about blood test results.
Line 220: “This may also partially explain the higher use of alcohol and smoking”. This might be the case. But the authors do have the data. Why not check if this explanation is true? A comparison stratified by sex would be a simple way to check this hypothesis.
Line 260ff: “there was a significant different [sic!] between the pesticide using and organic farmers for BMI but not the % body fat or waist circumference because these two groups have similar height and weight but different muscle weight and body fat distribution“: I do not understand this reasoning. If height and weight are similar in the two groups, how then can BMI differ significantly? And in fact sprayers have higher BMI. So to explain that they still have the same body fat we would have to assume that they have higher muscle mass. But we do learn that organic farmers do more sports and have a more labour intensive job. This is counterintuitive!
Author Response
Reviewer 1
This is a cross-sectional study comparing 2 groups of Thai farmers: pesticide sprayers versus organic farmers. The study investigates differences in metabolic parameters (glucose and lipid levels, blood pressure, BMI, body fat distribution). This is a valid approach and the authors promise that they will follow up the sample. So although this is a cross-sectional study there are further (cohort study) results to be expected in the future.
The authors investigate a long list of outcomes defined as binary (normal versus pathological value). I wonder why they not also investigated the effects on a continuous basis as originally they do have continuous data. I do agree that an increase in the frequency of pathological values is relevant. But often with continuous values the study power is increased and it is also interesting to quantify the size of an effect. Studying and discussing only binary values also led the authors to some mistakes. E.g. on page 9, line 271/272 they claim: “Pesticide using farmers had significantly higher levels of triglycerides, TC and LDL but lower HDL than organic farmers.” The latter statement is not true. Pesticide farmers do have a lower number of pathological values of HDL but that would mean they have higher HDL levels.
Thank you for pointing out the HDL error, we have corrected the interpretation. This analysis presents the baseline data for these two groups of farmers: those who currently use pesticides and organic farmers. The goal of this paper is to examine whether, at baseline, there was a differential risk for pathological levels of some of biomarkers that contribute to metabolic syndrome. We intend to look at the continuous data on these biomarkers in our longitudinal analysis of these subjects.
The 2 groups of farmers differ in many instances like age, sex, education, and alcohol and tobacco consumption. The authors do correct for these confounders in their final model. In general sprayers display unhealthier features (alcohol and tobacco, exercise) and have more (additional) indicators of pesticide exposure (years of use, distance to farm, use at home). One would assume that these factors would by their own increase the risk of metabolic pathologies. So when controlling for these factors we would expect the remaining OR to be lower. But in fact table 5 demonstrates stronger effects after adjustment. The reader might wonder which factor was driving this effect of adjustment. It is mainly age.
The final models included risk factors that were significantly associated with the metabolic outcomes in a number of univariate models. Thus, we did not control for education and distance to farm which were not found to be significant in univariate models. We modified the models so the historical pesticide use variable includes 5 categories: never used pesticides, 1-10 years, 11-20 years, 21-30 years and more than 30 years.
I would expect pesticide exposure to affect metabolic pathologies through chronic exposure. Therefore, I would expect stronger effect with duration of exposure compared to current exposure which is the only exposure indicator for which results are reported. But the authors even included years of exposure in their models as a confounder. It would be interesting to learn about the impact of this exposure indicator as well.
We have tried to control with years of pesticides used and without years of pesticides used; the effects of were stronger in most of parameters when controlling with history of pesticide used as in Table 1 below.
Table 1 Comparison of health outcomes between chemical use (n=216) and organic farmers (n=231) using generalized linear model for probability of abnormal blood test result.
Health outcome | Chemical and organic farmers Adjusteda RR (95%CI) | p-value | Chemical and organic farmers Adjustedb RR (95%CI) | p-value |
BMI | 2.08(1. 38-3.11) | <0.001 | 1.83(1.20-2.78) | 0.005 |
Waist circumference | 1.66(1.16-2.37) | 0.006 | 1.69(1.13-2.51) | 0.010 |
% body fat | 1.30(1.07-1.57) | 0.008 | 1.31(1.05-1.64) | 0.018 |
Triglyceride | 1.40(0.97-2.04) | 0.076 | 1.51(1.01-2.27) | 0.045 |
Cholesterol | 2.00(1.580-2.53) | <0.001 | 2.20(1.69-2.86) | <0.001 |
HDL | 0.81(0.71-0.93) | 0.002 | 0.83 (0.73-0.95) | 0.008 |
LDL | 1.29(1.12-1.49) | 0.001 | 1.34 (1.14-1.57) | <0.001 |
Blood glucose | 1.50 (0.67-3.35) | 0.321 | 1.54(0.67-3.53) | 0.309 |
Blood pressure | 1.59(1.02-2.50) | 0.043 | 1.54(0.98-2.42) | 0.064 |
Metabolic syndrome | 1.36(0.92-2.02) | 0.121 | 1.42(0.94-2.13) | 0.094 |
a Adjusted for sex, age, smoking, drinking alcohol, eating fruits and vegetables less than half a kilogram 5-7days/week, heavy exercise, and insecticide use at home
b Adjusted for the same parameters including history of pesticide used
Minor details:
The authors describe the sprayers by several different terms: pesticide exposed farmers, chemical use farmers, chemical farmers,…. The latter term is problematic. But also “chemical use” is not very nice because in the end “everything” is a chemical, even water (H2O). Why not define a single term and then use it throughout the paper. I now used “sprayers” but the authors also could use “conventional farmers” or whatever term they prefer.
Thank you for this comment. We have changed the term to conventional farmers throughout.
Tables are good to report many figures. It is not necessary to report all the figures again in the text. For example the findings reported in table 1 are again reported in detail on page 4, lines 118ff. It would suffice only to report the main highlights. The details on second jobs and the types of plants grown could also go into the table instead of the text. Similar considerations apply for the risk factors (table 2). Especially the description of drinking water sources gets confusing and does not really seem necessary.
Thank you for this suggestion, we have cut the descriptions of table results in the text and put the details on the types of plants grown into the table. We have maintained the drinking water source categories in the table as drinking water may be another source of pesticide exposure.
Table 4 and table 5 (unadjusted part) are redundant.
We have taken out the crude RR (95%CI).
Language needs improvement. One example (apart from the term “chemical farmer”) is use of plural/singular forms, e.g. lines 18 (exposure – factors), 19 (there is only one metabolic syndrome), 123-124 (hours was). Some terms are not very usual like “expense adequacy” (table 1): why not “income”? When you classify farmers into age groups (line 119ff) you do not show that the sprayers are “younger” but that there are “more young farmers” among that group. I do not understand why raising animals or growing plants (lines 127f) is not a farmer’s job.
We have renamed the variable “income” in Table 1. The organic farmers have a higher and statistically significantly different mean age (53 vs 50) and higher max age in the group (79) as well as fewer subjects < 35 and more subjects > 60. We took out the raising organic animals. We have had the manuscript reviewed again by a native English speaker to address the other points.
Line 105: I am sure this was not a Poisson regression but most likely a logistic regression because the outcome variables are binary.
This is generalized linear model using a Poisson loglinear function. Due to the high percentage of the population with the health outcomes, it did not meet the criteria of a rare outcome (logistic regression OR).
Line 176: two times organic farmers.
The grammar was corrected.
I would not classify “no pesticide use ever” and up to 10 years of pesticide use in the same category (table 2).
We have classified pesticide use history into 0 pesticide use, 1-10 years, 11-20 years, 21-30 years and more than 30 years.
Line 58ff: (also stated later in the discussion): it is simply not true that this is the first study comparing exposed to unexposed farmers. Even this journal only recently published 2 papers from Dom Rep where organic and conventional farmers were compared. Strengths and weaknesses of a paper are usually discussed in the discussion section, not advertised in the introduction.
We have removed the statement of study strengths and as well as findings of other studies on organic and conventional farmers to the discussion section.
Line 72: what means “second highest organic farming area”? Percentage or area size?
This refers to the size of the area.
Line 74: “males or female farmers“: this text could be improved.
This was edited.
Description of table 5: Odds ratios of pathological values in conventional farmers compared to organic farmers. If you only write “comparison between” you could also mean the ORs for organic farming (compared to the other group). In the methods section the authors claimed they did a Poisson regression. Here they state it is a GLM which indeed would allow for several families of models. “Poisson” would be wrong and “GLM” is not very informative. And, by the way, the results are not exclusively about blood test results.
We used GLM with a Poisson loglinear function because of the high percentage of the population with the outcome. A number of authors have recommended using Poisson models to estimate the RR due to the increasing differential between the OR and RR when the outcome incidence exceeds 10%
1. McNutt LA, Wu C, Xue X, Hafner JP. Estimating the Relative Risk in Cohort Studies and Clinical Trials of Common Outcomes. Am J Epidemiol 2003; 157(10):940-3.
2. Zou G. A Modified Poisson Regression Approach to Prospective Studies with Binary Data. Am J Epidemiol 2004; 159(7):702-6.
3. Sander Greenland , Model-based Estimation of Relative Risks and Other Epidemiologic Measures in Studies of Common Outcomes in Case-Control Studies, American Journal of Epidemiology 2004;160:301-305
Line 220: “This may also partially explain the higher use of alcohol and smoking”. This might be the case. But the authors do have the data. Why not check if this explanation is true? A comparison stratified by sex would be a simple way to check this hypothesis.
We have examined with stratified Chi-Square analysis and found that smoking and drinking alcohol are significant by sex (p<0.001 and <0.001) in both conventional and organic farmers, respectively
Line 260ff: “there was a significant different [sic!] between the pesticide using and organic farmers for BMI but not the % body fat or waist circumference because these two groups have similar height and weight but different muscle weight and body fat distribution“: I do not understand this reasoning. If height and weight are similar in the two groups, how then can BMI differ significantly? And in fact sprayers have higher BMI. So to explain that they still have the same body fat we would have to assume that they have higher muscle mass. But we do learn that organic farmers do more sports and have a more labour intensive job. This is counterintuitive!
Thank you for this comment. We have explained that when comparing conventional and organic farmers there was a significant difference in the % with abnormal BMI while the percent of those with abnormal % fat and abnormal waist circumference were not significantly different. This may be because the criteria for normal and abnormal % body fat and waist circumference take into account gender, while the criteria for abnormal BMI does not account for gender. Because there are more men in the conventional farmer group and they may have more muscle mass their BMI may be higher and fall into the “abnormal” range even though their % body fat or waist circumference do not.

Reviewer 2 Report
This paper addresses an important topic- the differences in both risk factors and health outcomes for chemical and organic farmers in Thailand. I suggest a few changes to make the manuscript stronger:
1. Editing for grammar especially punctuation and spacing.
2. A little more information in the introduction about what is known about the impact of pesticides on metabolic diseases.
3. More information on how recruitment was done, how data collection was done, who did the data collection, if there was remuneration for participants is needed.
4. The scale used to assess stress is not named or defined.
5. How do education, level, marital status, and "expense adequacy" related to risk factors? These social determinants of health are not really discussed.
6. A citation is needed in the discussion about rates of smoking and drinking in males versus females in Thailand.
7. What is a "near term impact"?
8. Do participants get their individual health results?
9. Are there plans to reduce risk behaviors? To intervene?
10. The conclusion should be more than one sentence.
Author Response
Reviewer 2
This paper addresses an important topic- the differences in both risk factors and health outcomes for chemical and organic farmers in Thailand. I suggest a few changes to make the manuscript stronger:
1. Editing for grammar especially punctuation and spacing.
We have had the manuscript reviewed again by a native English speaker
2. A little more information in the introduction about what is known about the impact of pesticides on metabolic diseases.
We have added a sentence describing research on pesticides and metabolic diseases to the introduction.
3. More information on how recruitment was done, how data collection was done, who did the data collection, if there was remuneration for participants is needed.
Thank you. We have added more detail on the data collection process.
4. The scale used to assess stress is not named or defined.
The questionnaire was from the Ministry of Health, Department of Mental Health Stress questionnaire. We have added a reference to this in the paper.
Department of Mental Health. Stress evaluation (In Thai). Available online: https://www.dmh.go.th/test/download/view.asp?id=18 (accessed 11 November 2018).
5. How do education, level, marital status, and "expense adequacy" related to risk factors? These social determinants of health are not really discussed.
The goal of this paper is not to examine how these demographic/social determinants relate to each other. However, we did want to report on the similarities or differences in basic demographic/social determinant variables between the two farming groups to see if they might be helpful in explaining differences we found in health outcomes between the two farming groups.
6. A citation is needed in the discussion about rates of smoking and drinking in males versus females in Thailand.
This was added in the discussion.
7. What is a "near term impact"?
The two farming groups represent current pesticide use versus no current pesticide use. Thus a significant association between farming group and health outcome after controlling for other risk factors, including historical pesticide use, suggests that current pesticide use (near term exposure) may be an important risk factor in altering metabolic biomarkers. We have tried to clarify this statement in the discussion.
8. Do participants get their individual health results?
Yes, all individual health check-up results were explained and provided to the subjects during a subsequent visit where ways to reduce risky behaviors was also addressed
9. Are there plans to reduce risk behaviors? To intervene?
The main study is a longitudinal study over a two-year period. We do not have an intervention component. However, the subjects and the local clinic are given the results and the subjects are referred to doctors if health results warrant follow up. So, little by little the subjects are realizing the health impacts of their work and lifestyle choices and what can be done to reduce their higher risk behaviors.
10. The conclusion should be more than one sentence.
The conclusion was revised.

Reviewer 3 Report
This study provides a detailed overview of a sample of 436 farmers in Thailand, with respect to current and past pesticide use and current components of metabolic syndrome (BMI, Waist Circumference, adiposity and cardiometabolic characteristics). The sample was designed to be free of diagnosed diabetes, high blood pressure, heart or thyroid disease. Results reflect a cross-sectional analysis of baseline data, as a foundation for future prospective analyses. Findings suggest that after adjusting for certain confounders, current use of pesticides in farmers is associated with higher cardiometabolic risk.
The subject matter is important and with some exceptions, clearly presented. There are some missing pieces, however, that may influence the interpretation of the findings. First, and foremost, the authors description of the sample needs to include some description of the response rate, and so does not allow the reader to interpret potential selection/response bias. Individuals with diagnosed disease were not eligible for the sample. Were there any differences in the number surveyed (prior to sample selection) by type of farming (organic or not).
A few specific comments follow:
Methods
(1) See main comments – a more thorough description of sampling frame and response rate is needed.
(2) The organic farmers appear to be older than the other farmers, with higher educational attainment, and were less likely to live near the farm. The models adjusted for several potential confounders, but it is unclear how these were identified and whether educational status was considered. As residence (rural or not) and education may explain access to healthcare, these may be important covariates. Model building strategy needs to be explained.
Results
(1) It might be helpful to see comparisons in Tables 1-3 stratified by age and sex
(2) The text explaining Table 2 is confusing when it comes to describing pesticide history. In lines 155-156, the text states that ~13% of organic farmers never worked in farm spraying pesticides, while only 14% reported ever spraying pesticides on their farms. This is not helped by the table, which groups never use into 0-10 years. Please add to the table “never used pesticides on a farm” and a show the average years and range used pesticides
(3) For models shown in Table 5, is history of pesticide use the dichotomous “year” variable. Is this sufficient to capture the variability?
(4) Consider sensitivity analyses, stratified by sex and/or age.
Discussion
(1) Please discuss generalizability of these findings and potential selection bias
(2) In lines 261-262, it seems strange that you are saying the two groups have similar height and weight, when their BMI is different (since BMI is a function of height and weight). Higher muscle, would imply higher BMI for a similar waist circumference? On the other hand, adiposity was not very different, while BMI was higher in the pesticide users. Is this also due to age and sex (two other major factors). Younger adults and men may have higher BMI due to muscle mass.
(3) The authors may wan to describe the different crops and types of pesticides used historically. Also, not all have the same actions, even if many have endocrine effects. It may be advisable to consider limiting discussion of specific pesticides and mechanisms. Or else, provide some additional context for this speculation, such as known distribution of pesticide use for different crops (or anything more specific you may have collected on this sample).
Author Response
Reviewer 3
Comments and Suggestions for Authors
This study provides a detailed overview of a sample of 436 farmers in Thailand, with respect to current and past pesticide use and current components of metabolic syndrome (BMI, Waist Circumference, adiposity and cardiometabolic characteristics). The sample was designed to be free of diagnosed diabetes, high blood pressure, heart or thyroid disease. Results reflect a cross-sectional analysis of baseline data, as a foundation for future prospective analyses. Findings suggest that after adjusting for certain confounders, current use of pesticides in farmers is associated with higher cardiometabolic risk.
The subject matter is important and with some exceptions, clearly presented. There are some missing pieces, however, that may influence the interpretation of the findings. First, and foremost, the authors description of the sample needs to include some description of the response rate, and so does not allow the reader to interpret potential selection/response bias. Individuals with diagnosed disease were not eligible for the sample. Were there any differences in the number surveyed (prior to sample selection) by type of farming (organic or not).
A few specific comments follow:
Methods
(1) See main comments – a more thorough description of sampling frame and response rate is needed.
This was a convenience sampling framework where we recruited subjects from the target agricultural areas who met our inclusion and exclusion criteria. There were243 conventional farmers and 235 organic farmers who agreed to participate. From the initial recruitment to the first physical health checkup 8 months later we lost some subjects, resulting in a 214 conventional farmers and 222 organic farmers for a response rate of 88.1 and 94.5%, respectively.
(2) The organic farmers appear to be older than the other farmers, with higher educational attainment, and were less likely to live near the farm. The models adjusted for several potential confounders, but it is unclear how these were identified and whether educational status was considered. As residence (rural or not) and education may explain access to healthcare, these may be important covariates. Model building strategy needs to be explained.
All demographic and risk factor variables were examined as covariates in univariate models with each of the health outcome parameters. If the variable showed significance in a number of univariate analyses, it was included in the final model. Thailand has a universal health care scheme and all sub districts (villages) have primary care units. All subjects in this study were from rural areas since they are farmers.
Results
(1) It might be helpful to see comparisons in Tables 1-3 stratified by age and sex
Thank you for your comment however the goal of this paper is not to examine how these demographic/social determinants relate to each other, rather we are focused on the role of pesticides in these health outcomes. However, we did want to report on the similarities or differences in basic demographic/social determinant variables between the two farming groups to see if they might be helpful in explaining differences we found in health outcomes between the two farming groups.
(2) The text explaining Table 2 is confusing when it comes to describing pesticide history. In lines 155-156, the text states that ~13% of organic farmers never worked in farm spraying pesticides, while only 14% reported ever spraying pesticides on their farms. This is not helped by the table, which groups never use into 0-10 years. Please add to the table “never used pesticides on a farm” and a show the average years and range used pesticides.
Thank you for catching this mistake. We have corrected it and added the average and range of years used pesticides to the table.
(3) For models shown in Table 5, is history of pesticide use the dichotomous “year” variable. Is this sufficient to capture the variability?
The model uses a categorical variable with 5 categories: no pesticide use, 1-10 years,11-20 years, 21-30 years and 21->30 years.
(4) Consider sensitivity analyses, stratified by sex and/or age.
The importance of sex and age is not a primary interest of this paper. However, we do control for them in our final model.
Discussion
(1) Please discuss generalizability of these findings and potential selection bias
The potential selection bias may come from those who think that they are healthy, so they would like to participate in the study compared to those who afraid of having an abnormal physical health outcome, so do not want to participate in the study.
We believe the outcomes are generalizable in that it suggests that poorer health outcomes in metabolic biomarkers are associated with current use of pesticides, not just cumulative historical exposures. We have included comments on generalizability in the discussion.
(2) In lines 261-262, it seems strange that you are saying the two groups have similar height and weight, when their BMI is different (since BMI is a function of height and weight). Higher muscle, would imply higher BMI for a similar waist circumference? On the other hand, adiposity was not very different, while BMI was higher in the pesticide users. Is this also due to age and sex (two other major factors). Younger adults and men may have higher BMI due to muscle mass.
Thank you for this comment. We have explained that when comparing conventional and organic farmers there was a significant difference in the % with abnormal BMI while the percent of those with abnormal % fat and abnormal waist circumference were not significantly different. This may be because the criteria for normal and abnormal % body fat and waist circumference take into account gender, while the criteria for abnormal BMI does not account for gender. Because there are more men in the conventional farmer group and they may have more muscle mass; their BMI may be higher and fall into the “abnormal” range even though their % body fat or waist circumference do not.
(3) The authors may want to describe the different crops and types of pesticides used historically. Also, not all have the same actions, even if many have endocrine effects. It may be advisable to consider limiting discussion of specific pesticides and mechanisms. Or else, provide some additional context for this speculation, such as known distribution of pesticide use for different crops (or anything more specific you may have collected on this sample).
We provide the list of pesticides currently used by conventional farmers.
